# Enhanced Biological Nitrate Removal from Groundwater in Humid Tropical Regions Using Corn Cob-Based Permeable Reactive Barriers: A Case Study from Panama

Graciela Cecilia Sánchez Hidalgo [1], Maria De Los Ángeles Ortega [1,2,3,4] and Euclides Deago [3,4,5,6,*]

1 Research Group—Iniciativa de Integración de Tecnologías para el Desarrollo de Soluciones Ingenieriles (I2TEDSI), Faculty of Mechanical Engineering, Universidad Tecnológica de Panamá, El Dorado, Panama City 0819-07289, Panama; graciela.sanchez@utp.ac.pa (G.C.S.H.); maria.ortega@utp.ac.pa (M.D.L.Á.O.)

2 Research Group in Design, Manufacturing and Materials (DM+M), Faculty of Mechanical Engineering, Universidad Tecnológica de Panamá, Panama City 0819-07289, Panama

3 Sistema Nacional de Investigación (SNI), Clayton City of Knowledge Edf. 205, Panama City 0816-02852, Panama

4 Centro de Estudios Multidisciplinarios en Ciencias, Ingeniería y Tecnología (CEMCIT-AIP), Panama City 0819-07289, Panama

5 Centro de Investigaciones Hidráulicas e Hidrotécnicas (CIHH), Universidad Tecnológica de Panamá, Panama City 0819-07289, Panama

6 Research Group—Nitrato y Medio Ambiente, Faculty of Civil Engineering, Universidad Tecnológica de Panamá, El Dorado, Panama City 0819-07289, Panama

* Correspondence: euclides.deago@utp.ac.pa

**Abstract:** Nitrate contamination in groundwater is a global concern due to its widespread presence and consequential social, environmental, and economic ramifications. This study investigates the efficacy of biological denitrification in a humid tropical setting, utilizing corn cob in batch and column tests to assess nitrate removal under varying conditions. Batch tests demonstrated the nitrate removal efficiencies of 93.14%, 91.58%, 90.77%, and 98.74% for initial concentrations of $22.18 \pm 2.82$ mg/L, 27.3 mg/L, $69.1 \pm 1.2$ mg/L and $115.08 \pm 1.88$ mg/L, respectively. In the column test, the removal efficiency was 99.86%, 87.13%, and 74%, and the denitrification rate was 32.82, 53.43, and 83.53 mg $NO_3^-$-N/L d, for a hydraulic retention time (HRT) of 24 h, 16 h, and 7 h, respectively. Predominantly, nitrate removal occurred via biological denitrification, particularly favoring a 24 h HRT. The corn cob effectively removed high nitrate concentrations of up to 115 mg $NO_3^-$-N/L. Scanning electron microscopy and Fourier transform infrared spectroscopy revealed surface characteristic changes of the carbon source pre- and post-denitrification. This research sheds light on the potential of biological denitrification using corn cob in humid tropical environments, offering a promising avenue for addressing nitrate contamination challenges in groundwater systems.

**Keywords:** nitrate; corn cob; denitrification; groundwater; permeable reactive barrier

## 1. Introduction

Groundwater pollution remains a critical issue worldwide [1,2]. Nitrate ($NO_3^-$-N) has emerged as the most prevalent anthropogenic contaminant in groundwater globally, largely attributable to the excessive application of fertilizers and nitrogenous chemicals that seep into both surface and groundwater systems [3,4]. In 2020, the use of inorganic fertilizers in agriculture increased by 46% compared to 1990 figures, with nitrogen-based fertilizers making up 56% of this usage, followed by phosphorus and potassium fertilizers at 24% and 20%, respectively [5]. In water, $NO_3^-$-N is the most stable and mobile form of nitrogenous species. Thus, it tends to accumulate in aquifers, posing significant health risks to human populations and ecosystems [6–8]. In response, both the World Health

Organization (WHO) and the Food and Agriculture Organization of the United Nations (FAO) have established quality limits for drinking and irrigation water, with a limit of $NO_3^-$-N concentration of 50 mg/L and 22 mg/L, respectively [3,9,10].

An alternative for in situ polluted groundwater treatment is using permeable reactive barriers (PRBs) [11]. Over the past two decades, PRBs have seen a notable rise in popularity, increasingly replacing the traditional ex situ pump and treat methods due to their promising results and comprehensive approach to addressing underground pollution [12]. The application of PRBs using organic materials as reactive materials has improved the growth and activity of microorganisms, facilitating the degradation of contaminants that other reactive materials, such as zero-valent iron (ZVI), could not effectively treat [11].

PRBs use reactive materials like organic compounds, polymers, and agricultural residues to treat nitrate-contaminated groundwater. These materials are a carbon source in denitrification [13]. During biological denitrification, nitrate is converted into nitrogen gas by denitrifying bacteria that use the carbon source for electron donation and as a biofilm carrier (Figure 1) [14]. Consequently, agricultural waste has emerged as a particularly effective natural organic solid substrate (NSOS) for nitrate removal [14]. It provides a readily available and abundant source of organic carbon and electron donors, making it an advantageous strategy for this purpose [15].

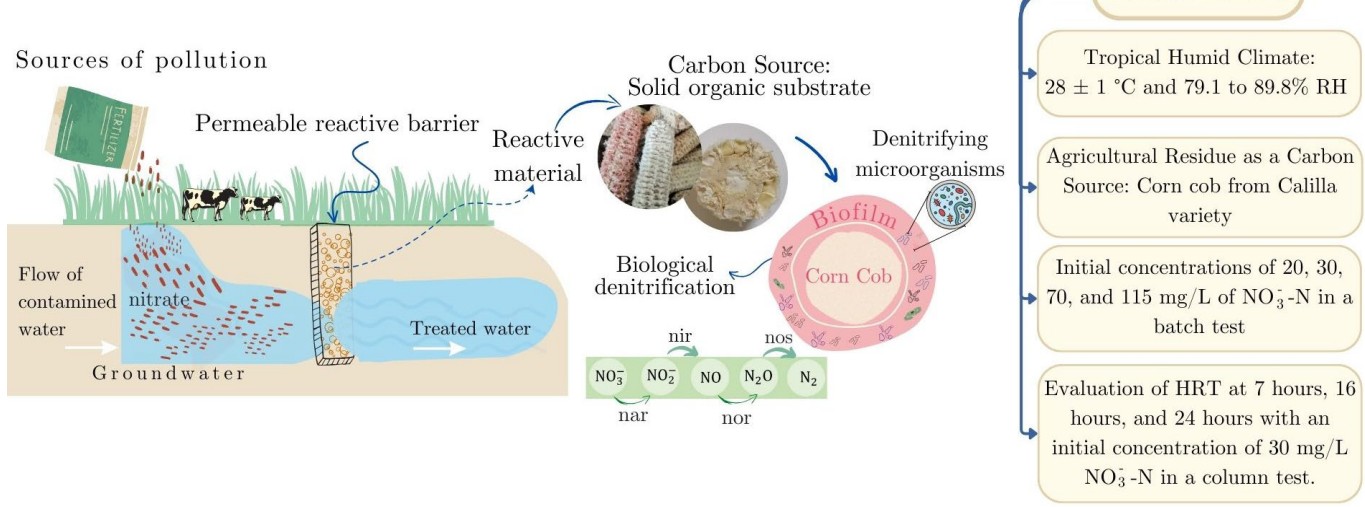

**Figure 1.** Scheme of a permeable in situ reactive barrier for removing nitrate in groundwater.

Among the agricultural residues previously studied in the literature for their potential in nitrate removal are wheat straw [16], mulch [17], luffa [18], canal straw [19], and corn cob [20]. For instance, Xu et al. [21] research demonstrated that corn cobs could remove over 90% of nitrate from wastewater, showcasing their potential as an economical and effective carbon source for denitrification. This finding is supported by Liu et al. [20], who observed effective nitrate removal using corn cobs in groundwater at 16 °C in China. Similarly, Li et al. [22] recognized corn cob as a viable carbon source for denitrification processes. Additionally, Yuan et al. [23] reported that adding 1% corn cobs to constructed wetlands significantly improved denitrification efficiency, increasing $NO_3^-$-N removal from 19% to 71.9%.

In Panama, corn ranks seventh among the most-produced crops, with the 2021 harvest yielding 135,425 tons [24]. This increase in production has resulted in a substantial amount of agricultural waste, including corn cobs [25]. Such an abundance of waste presents an opportunity to use this resource for environmental remediation. Therefore, this study aimed to explore the potential of corn cobs for removing nitrate from contaminated groundwater at various concentrations through batch testing and simulating a permeable reactive barrier in a column test under tropical environmental conditions.

Panama faces significant challenges related to water supply and quality [26]. Furthermore, there is a lack of updated monitoring systems for groundwater quality [27]. Rural areas where community water supply and sanitation systems are often inadequate are more vulnerable to contamination [28]. This study aligns with achieving the Sustainable Development Goals (SDGs), specifically SDG 6, which focuses on ensuring clean water and sanitation access. By addressing these challenges, this research aims to propose solutions that enhance the quality of life in affected communities [29].

*State of the Art*

The nitrate removal efficiency of corn cob as a carbon source has been extensively investigated. Liu et al. [20] explored the effectiveness of corn cob combined with 0.1 g of denitrifying bacteria at $16 \pm 1$ °C over 15 days. Their results indicated high nitrate removal efficiencies in a column test with a 24 h hydraulic retention time (HRT), achieving 99.8%, 97.8%, 94.3%, and 85.9% for initial nitrate concentrations of 20, 40, 60, and 80 mg/L, respectively. Similarly, Xie et al. [30] used a mixture of polyhydroxyalkanoates (PHA), straw, corn cob, and bagasse in a permeable reactive barrier system at a laboratory scale. They reported a decrease in nitrate removal efficiency from 99.8% to 73.1% as the nitrate concentration increased from 40 to 60 mg/L in groundwater.

In the study by Ling et al. [31], corn cob was found to remove 5.12 mg of nitrate, attributed to its rough surface and high content of hemicellulose and cellulose, surpassing the performance of wheat straw, which only removed 1.14 mg due to its smoother surface for denitrifying bacteria, making it less conducive to denitrifying bacteria adhesion and activity. Similarly, Xu et al. [21] reported that the denitrification rate increased to 203 mg $NO_3^-$-N/L/day when using corn cob, with flow rates ranging from 8.5 to 153 L/day.

Low nitrite and ammonium concentrations may indicate that nitrate reduction occurred through complete denitrification. On the other hand, under conditions with limited electron availability, an increase in ammonium could occur due to processes such as dissimilatory nitrate reduction to ammonium (DNRA) instead of its complete transformation into nitrogen gas. Increased ammonium can be considered a contamination risk [32,33].

Previous studies have reported nitrite and ammonium concentrations below 0.5 mg/L, indicating complete denitrification and not processes like DNRA when organic substrates were used as electron donors in the denitrification process. In this regard, other studies reported nitrite concentrations where denitrification was evaluated using an organic substrate [34]. For example, results reported by Guo et al. [16] indicated nitrite concentrations between 0.50 mg/L and 0.60 mg/L when using wheat straw, suggesting complete denitrification due to these low nitrite concentrations. Nitrite concentrations can vary depending on the carbon source, environmental conditions, and initial nitrate concentration. For instance, Feng et al. [35] reported nitrite concentrations of 0.274 mg/L in sugarcane straw and 0.217 mg/L in corn cob. Similarly, Xie et al. [30] reported nitrite concentrations fluctuating between 0 and 1 mg/L when using the mixture of PHA, straw, corn cob, and bagasse. Therefore, this study evaluated corn cob as a carbon source for denitrification due to its accessibility and affordability for use in permeable reactive barriers under humid climatic conditions such as those in Panama.

## 2. Materials and Methods

In this research, an experimental methodology was employed to explore the potential of corn cob as a carbon source for removing nitrate in groundwater. Figure 2 schematically illustrates the methodological sequence used in the study.

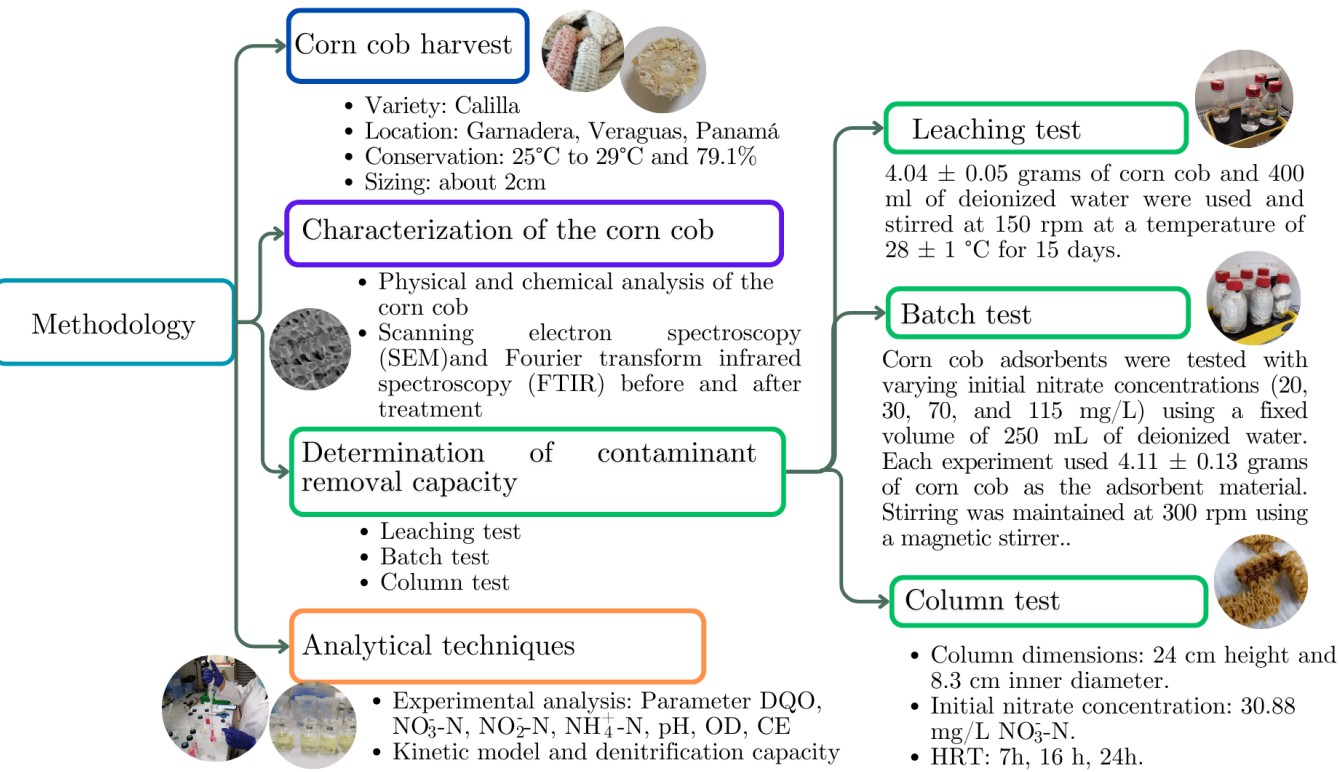

**Figure 2.** Outline of the methodology used.

### 2.1. Harvesting the Corn Cob

Corn cobs of the calilla variety were collected at the Garnadera site, Atalaya Cabecera, Veraguas (Coordinates: E-505561.80 m, N-888013.97 m, UTM, WGS-84) (1TRF-97) [36]. The corn cocoons were harvested dry, with grains manually extracted. The selection was based on a visual inspection, ensuring that no visible damage was present on the ears. Subsequently, the cobs were transported to the Technological University of Panama in Panama City. They were stored at room temperature, maintained between 25 °C and 29 °C with 79.1% humidity, for a period of approximately six months. For experimental use, the cobs were sized to approximately 2 cm [21].

### 2.2. Physical and Chemical Characterization of Corn Cobs

To determine the composition of the corn cob fiber, samples were processed in a mechanical convection oven at 60 °C for 72 h using a Yamato DKN810 (New York, NY, USA). After drying, they were ground to a particle size of 1 mm using equipment from Restsch GmbH & Co. (Haan, Germany). The chemical composition of the corn cob was obtained via digestion with HCl 1:1 from ash for the elements Ca (calcium), Zn (zinc), Mg (magnesium), Fe (iron), K (potassium), Cu (copper), Na (sodium), Mn (manganese), with atomic absorption spectrophotometry with an individual hollow cathode lamp. Phosphorus was determined via digestion with HCl 1:1 from ash with molybdo-ammonium vanadate (visible spectrophotometry).

Several physical parameters of the corn cob were determined: biodegradable fraction of the material, which was obtained from the equation proposed by Chandler and Jewell [37] and following the procedures of the Standard Method [38]; hydraulic conductivity of the corn cob, which was obtained from the constant head methodology (ASTM D2434) [39]; and the porosity of the columns, which was determined gravimetrically from the amount of water that was introduced into the solid [4]. The morphology of the corn cob before and after treatment was observed by scanning electron spectroscopy (SEM) [35,40]. Samples were randomly taken for observation, and the carbon distribution on the surface of the corn cob was analyzed using energy dispersive X-ray spectroscopy (EDS) SEM and a Zeiss

Evo 40 VP scanning electron microscope. This analysis was performed in the Electron and Confocal Microscopy laboratory at the Smithsonian Tropical Research Institute of Panama.

Fourier transform infrared spectroscopy (FTIR) was used to identify the functional groups in corn cobs. Initially, the cobs were dried at a controlled temperature of $(105 \pm 1)$ °C for 24 h. After drying, they were ground to a particle size of 1 mm using a Thomas Model 4 Wiley® Mill. A 13 mm disk was prepared using the potassium bromide (KBr) pellet method for the infrared analysis. This involved compressing 300 mg of KBr and 10 mg of the sample under a hydraulic press at a pressure of 8 to 10 tons [41]. The samples were then analyzed using an Agilent Cary 660 FTIR spectrometer (Agilent Technologies, Santa Clara, CA, USA) The spectral resolution was set at 4 cm$^{-1}$, and the data were collected over eight scans.

### 2.3. Determination of Nitrate Removal Capacity

2.3.1. Leaching Test

The leaching test assessed the release of soluble components, including nitrogenous compounds and cations. This test was performed in triplicate using 500 mL glass bottles previously sterilized. Each reactor contained $4.04 \pm 0.05$ g of corn cob, which was disinfected using ultraviolet light (365 nm) to prevent microbial interference or cross-contamination [19]. To each reactor, 400 mL of sterilized distilled water was added [22]. The reactors were agitated at 150 rpm at $28 \pm 1$ °C for 15 days. Total nitrogen was quantified by summing the concentrations of nitrate, nitrite, and ammonium [18]. Additionally, the organic carbon content of the corn cob was measured in terms of the chemical oxygen demand (COD) to evaluate its organic carbon capacity [42].

2.3.2. Batch Test

In the batch test, contaminated groundwater was simulated using distilled water enriched with potassium nitrate fertilizer (Ferti-K Potassium Nitrate 13-0-46), which contains 13.4% total nitrogen and has a pH of 8. This setup was chosen to mimic environmental conditions where nitrogen fertilizers are a primary source of nitrate contamination in groundwater [3]. A preliminary test was carried out to rule out significant differences between the fertilizer and the analytical chemical ($KNO_3$). A $t$-test for independent samples was performed between the batch groups enriched with the fertilizer and those with the potassium nitrate-based analytical chemical. The results showed no significant differences in the nitrate concentrations of the groups, as indicated by a $p$-value with a significance level of 0.05. Consequently, the fertilizer was used for subsequent experiments.

Experiments were conducted using corn cob in distilled water to assess the impact of initial nitrate concentrations on removal efficiency. Four initial nitrate concentrations ($NO_3{}^-$-N) were tested: 20, 30, 70, and 115 mg/L, each with a volume of 250 mL water and $4.11 \pm 0.13$ g of corn cob. The experiments were organized into two groups: the first group with the initial concentrations of 20 and 115 mg/L underwent an acclimatization period of 5 days, while the second group with concentrations of 30 and 70 mg/L acclimatized for four days. The reactors were agitated on a magnetic stirrer at 300 rpm at room temperature for 14 and 12 days, respectively. To maintain anoxic conditions, the reactors were purged with nitrogen gas for 3 to 5 min [19]. Additionally, 3 mg/L of dipotassium phosphate buffer ($K_2HPO_4$) was added to keep the pH between 6 and 7.5. The reactors were kept in darkness to simulate aquifer conditions and prevent photosynthesis [4].

2.3.3. Column Test

Three acrylic columns, each 24 cm high and 8.3 cm in internal diameter, were used to simulate permeable reactive flow barriers operating under gravity flow without using pumps. The flow rates were controlled by regulating valves, as depicted in Figure 3. The experiment was conducted continuously, with adjustments to flow rates while maintaining the same reactive material. Hydraulic residence times (HRTs) were set at 7, 16, and 24 h. It was assumed that the groundwater flow through these permeable reactive barrier columns

followed a plug flow model, meaning that all the water within the column experienced the same hydraulic residence time, as calculated using Equation (1) [43].

$$Q = \frac{LAn}{HRT} \tag{1}$$

where $Q$ is the flow, $L$ is the linear distance, $An$ is the column's cross-sectional area, and $HRT$ is the hydraulic residence time.

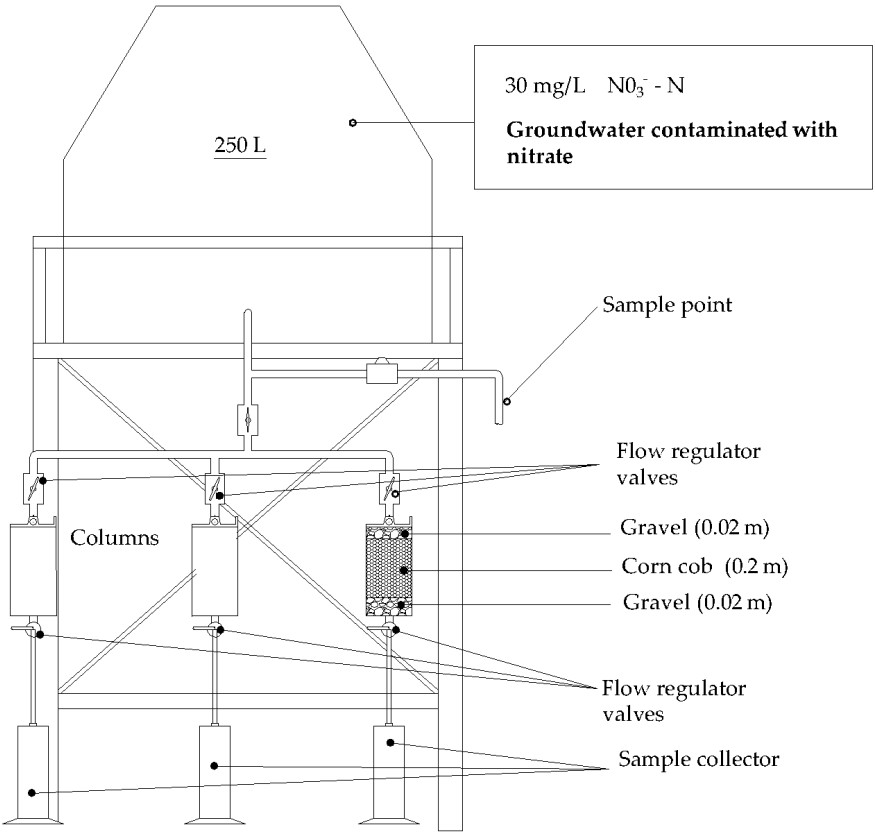

**Figure 3.** Test bench used for column testing, modified from [44].

Each column was segmented into three layers: a lower and upper gravel layer, each 2 cm thick, and a central reactive zone 20 cm high, containing 0.082 kg of corn cobs sized approximately 1.5 ± 0.5 cm. Mesh filters were placed between the layers to prevent material mixing. The groundwater used in the experiment was extracted from a well at the Technological University of Panama, Tocumen campus. In the feed tank of the column system (Figure 3), potassium nitrate fertilizer was added to achieve an initial nitrate concentration of 30 mg/L $NO_3^-$-N, supplementing the natural groundwater concentration of 0.97 mg/L $NO_3^-$N with a pH of 7.18. Additionally, the alkalinity of the groundwater was reported to be 264.29 mg/L [45]. Given the water's natural alkalinity, using buffer reagents in the column system was deemed unnecessary.

According to the HRT, water samples were taken at 48 and 56 h for each run. Additionally, to determine the parameters (pH, DO, $NO_3^-$-N, $NO_2^-$-N, $NH_4^+$-N, COD), the columns were covered to prevent light penetration and photosynthesis processes inside [4]. The average environmental conditions were 28 to 29 °C and relative humidity 86.8 to 89.8%.

### 2.4. Analytical Techniques

Experiments were conducted in triplicate, adhering to standard methods for water and wastewater characterization [38]. The samples were collected periodically and filtered by centrifugation at 6000 rpm for 10 min (Hettich® Universal 320®, Tuttlingen, Germany), then

filtered through filter paper (2235) 25–30 μm in diameter and 0.45-micron milli pore size. The pH, electrical conductivity, and dissolved oxygen were obtained using the HACH 40D multiparameter instrument. For the analysis of low and high nitrate ranges, the cadmium reduction method 8192 and cadmium reduction method 8039, respectively, were used. For nitrite and ammonium, the USEPA Diazotization method 8507 and salicylate method 8155, respectively, were employed. According to the Hach water analysis manual, the USEPA 8000 reactor digestion method determined the COD using the Hach DRB200 and DR/6000 reactor (DR6000 UV-VIS Spectrophotometer, Hach, Düsseldorf, Germany) [46].

Kinetic Model and Denitrification Rate

The reaction rate of the column assays was analyzed using kinetic models. Previous studies have demonstrated that denitrification with corn cobs follows a pseudo-first-order kinetic model [20]. This model suggests that the reaction rate primarily depends on the nitrate concentration. The reaction rate is described by Equation (2) [43].

$$-r = \frac{dC}{dt} = KC \tag{2}$$

The reaction rate ($r$) can be expressed in terms of concentration ($C$) of the reactant (nitrate), and the reaction rate constant ($k$) represents the rate at which nitrate degrades in time ($t$) and can be determined from the slopes of linear plots of concentration data versus time. The initial nitrate concentration in the groundwater enters the reactive material (Co) zone. On the other hand, the half-life ($t_{\frac{1}{2}}$), which is the time it takes for the nitrate concentration to reduce to half its initial value, is shown by Equation (3) [43].

$$t_{\frac{1}{2}} = \frac{\ln 2}{k} \tag{3}$$

The removal efficiency was calculated based on the difference between the initial and final nitrate concentrations [20]. The removal rate was subsequently determined using Equation (4) [21].

$$r_D = \frac{Q \cdot (C_0 - C_E)}{V} \tag{4}$$

where $r_D$ ((mg $NO_3^-$-N)/ L × d) is the denitrification rate, $C_0$ (mg/L) is the nitrate concentration in the influent, $C_E$ (mg/L) is the nitrate concentration in the effluent, $Q$ is the flow rate (L/d), and $V$ is the volume of corn cobs in the reactor (L).

Pearson's correlation was used for statistical analysis to evaluate the linear relationship between two variables. Furthermore, variance analysis was used to determine the significant differences in the obtained data influenced by the initial nitrate concentration and HRT.

## 3. Results and Discussion
### 3.1. Physical and Chemical Characterization of Corn Cob

The ash content of the corn cobs was found to be 1.98%, which is consistent with the findings reported by [47]. The composition analysis revealed that hemicellulose, cellulose, and lignin constituted 50.8%, 34%, and 6% of the material, respectively (see Table 1). These proportions align with previous studies [48]. Additionally, the volatile solid content was determined to be 84.47 ± 2.88%, leading to a biodegradable fraction of 0.63 ± 0.01, similar to the values reported for sugarcane straw [19]. The higher biodegradability content is attributed to the lower lignin content in the organic substrates. This biodegradable fraction is a critical criterion for selecting an organic substrate due to its potential to enhance microbial activity and, thus, improve contaminant removal [49].

**Table 1.** Physicochemical composition of corn cob.

| Parameter | Value |
|---|---|
| Ash (%) | 1.98 |
| Lignin (%) | 6 |
| Hemicellulose (%) | 50.8 |
| Cellulose (%) | 34 |
| Dry Matter % | 92.07 |
| Moisture % | 7.95 |
| Volatile Solids (%) | $84.47 \pm 2.88$ |
| Biodegradable Fraction | $0.63 \pm 0.01$ |
| Bulk Density (kg/m$^3$) | 139.45 |
| Porosity | 0.46 |
| Hydraulic Conductivity (m/s) | $5.69 \times 10^{-5}$ |
| Nitrogen (N) % | 0.07 |
| Phosphorus (P) % | 0.06 |
| Potassium (K) % | 1.98 |
| Calcium (Ca) % | 0.04 |
| Magnesium (Mg) % | 0.02 |
| Iron (Fe) mg/L = ppm | 360.17 |
| Copper (Cu) mg/L = ppm | 2.09 |
| Manganese (Mn) mg/L = ppm | 0 |
| Zinc (Zn) mg/L = ppm | 96.50 |
| **Sodium (Na) mg/L = ppm** | 9.31 |

The chemical composition of the corn cob, detailed in Table 1, revealed high contents of potassium (1.98%), nitrogen (0.07%), phosphorus (0.06%), calcium (0.04%), and magnesium (0.02%). For heavy metals, the analysis found iron at 360.17 mg/L, zinc at 96.50 mg/L, and copper at 2.09 mg/L. These values are comparable to those found in rice husks [50]. Notably, manganese was not detected in the corn cob. Another study reported a higher nitrogen content in corn cob ($0.340 \pm 0.002\%$) than observed in this research [51]. Such chemical analyses provide valuable insights into the composition of corn cob, elucidating its potential as a source of carbon and energy [52].

*3.2. Determination of Contaminant Removal Capacity Using Corn Cob*

3.2.1. Leaching Assay

The leaching characteristics of corn cob were analyzed before the denitrification experiments to assess the potential contamination risks from the leachate.

Table 2 presents the results after 15 days, highlighting high concentrations of potassium (K) and calcium (Ca), similar to the findings reported for corn cob leachate [21] and sugarcane straw [19]. These elements, including Ca, K, Mg, Na, Si, and P, are essential for microorganism viability enzymatic activity, and may also contribute to stabilizing microbial cell walls [53].

**Table 2.** Leaching analysis of corn cob.

| Parameter (µg/L) | K | Ca | S | Ti | Cr | Cl | Fe | Ni | Cu | Zn | Ga | Br | Ba |
|---|---|---|---|---|---|---|---|---|---|---|---|---|---|
| Leaching | 12,310 | 550 | 511 | No det. | 12.4 | 3730 | 86.6 | 7.7 | 11.7 | 13.2 | 100 | 18.5 | 17.2 |

Zinc (Zn), iron (Fe), magnesium (Mg), copper (Cu), and manganese (Mn) are crucial for enzyme activity and the proper functioning of biological processes [53]. These metallic elements also enhance denitrification rates by acting as active centers in the denitrification process [21]. The concentrations of Fe, barium (Ba), Zn, Cu, chromium (Cr), and nickel (Ni) in the leachate were found to be low compared to those reported by Yang et al. [53] and Guan et al. [42] This difference may be attributed to the unique characteristics of corn cob.

The results of the leaching analysis indicated low concentrations of iron, copper, zinc, and sodium in the corn cob (Table 2). These findings suggest that, although these elements are present in the corn cob (Table 1), their release into the environment through leaching is minimal. Therefore, they are unlikely to pose a significant risk to groundwater.

Throughout the leaching process of corn cob, the average nitrate concentration stood at $0.048 \pm 0.015$ mg/L, falling within a range of 0.029 to 0.067 mg/L over the 14-day experimental period. As for nitrite, its average concentration was $0.019 \pm 0.011$ mg/L, ranging between 0.005 and 0.032 mg/L. Ammonium exhibited a higher average concentration of $0.039 \pm 0.046$ mg/L, peaking at 0.097 mg/L, higher than nitrate and nitrite. These data indicate an initial release of total nitrogen with a gradual increase throughout the experiment, consistent with findings reported by Zang et al. [54]. The relatively low concentrations of nitrogen species correlate with the corn cob's total nitrogen content of 0.07%, as determined during chemical characterization. This suggests a minimal risk of secondary contamination during denitrification, supporting conclusions drawn in previous studies [18,42,55].

The average chemical oxygen demand (COD) concentration observed was $190.65 \pm 26.74$ mg/L, showing a diminishing trend over time with fluctuations ranging from 150.83 mg/L to 216.67 mg/L. These findings align with other studies using agricultural residues, such as wood chips, corn cobs, rice husks, corn straw, wheat straw, sugarcane straw [56], *Typha angustifolia* [57], and sugarcane straw [19]. These studies similarly reported an initial release of carbon followed by a gradual decrease. Additionally, the dissolved organic matter initially released by the corn cob facilitates microbial growth and biofilm formation, enhancing biodegradation, as Li et al. noted [58].

The average leaching potentials of nitrate, nitrite, ammonium, and total inorganic nitrogen were 0.048 mg/g, 0.019 mg/g, 0.039 mg/g, and 0.106 mg/g, respectively. Nitrogen released from corn cob primarily exists as nitrate, followed by ammonium and nitrite. However, the total nitrogen concentrations were comparatively lower than those observed in other reactive materials, such as mulch, compost [4], hazelnut shell [55], almond shell, and loofah sponge [4,18,55]. Carbon release amounted to 16.83 mg/g, contrasting with findings for almond shell and loofah sponge, which registered values of 6.32 mg/g and 21.95 mg/g, respectively [18]. Thus, corn cob as a reactive material demonstrates adequate release without accruing undesired compounds for groundwater remediation [54].

Leachate analysis revealed a pH of $5.87 \pm 0.12$, attributed to inorganic matter and mobile components within the biomass [52]. Electrical conductivity was measured at $76.56 \pm 2.87$ μS/cm, indicating dissolved salts in water, as identified in prior studies [47,52].

### 3.2.2. Batch Test

- Denitrification performance

The nitrate concentration decreased progressively with time for all initial nitrate concentrations (($NO_3^-$-N) 20, 30, 70, and 115 mg/L, respectively), as shown in Figure 4a. The nitrate removal efficiency was 93.14%, 91.58%, 90.77%, and 98.74% for initial concentrations of $22.18 \pm 2.82$ mg/L, 27.3 mg/L, $69.1 \pm 1.2$ mg/L, and $115.08 \pm 1.88$ mg/L, respectively (Figure 4b). The removal efficiency decreased with the increasing initial concentration value. However, it was greater than 90% leaching, as seen in Figure 4b. These results indicated that the corn cob provided a source of carbon and other nutrients for the denitrification process at different nitrate concentrations, allowing for the removal of nitrate effectively to remove nitrate from groundwater at high concentrations, such as 115 mg/L.

The efficiency of nitrate removal decreased as the initial nitrate concentration increased, consistent with prior findings, indicating a progressive enrichment of denitrifying bacteria and an associated enhancement in nitrate removal efficiency [30]. This phenomenon was attributed to the potential reduction in the carbon-to-nitrogen (C/N) ratio at higher initial nitrate concentrations, which results in decreased removal efficiency due to diminished carbon availability. Conversely, lower nitrate concentrations may fail to provide sufficient energy for denitrifying bacteria toward the conclusion of the reaction, thereby potentially compromising nitrate removal efficacy. In contrast to previous investigations by Liu

et al. [20] and Xie et al. [30], the present study employed potassium nitrate fertilizer, leveraged native denitrifying microorganisms inherent in the cob matrix, and capitalized on the naturally occurring groundwater conditions, specifically alkalinity, which were conducive to the proliferation of indigenous denitrifying microorganisms in the substrate. This facilitated the enzymatic hydrolysis of structurally available carbon, thereby enhancing nitrate removal efficacy, particularly at elevated initial nitrate concentrations.

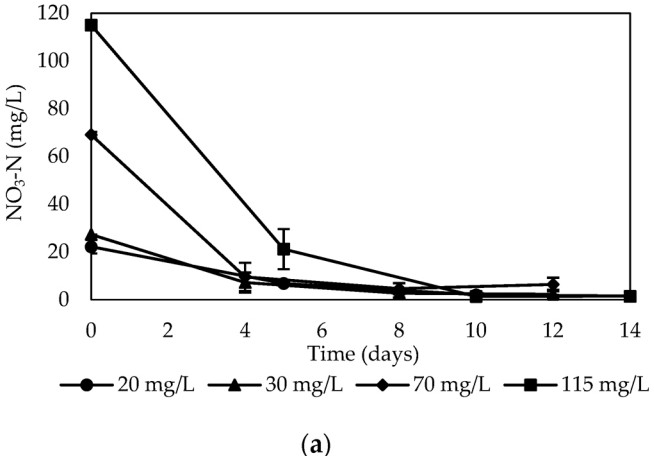
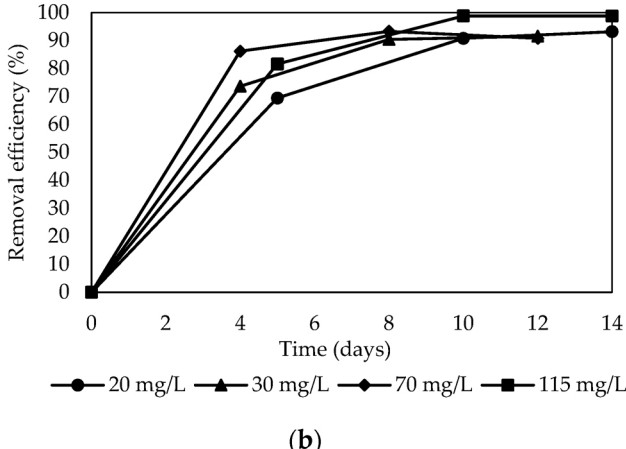

(a)  (b)

**Figure 4.** Denitrification performance using corn cob as reactive material: (**a**) behavior of nitrate removal; (**b**) nitrate removal efficiency.

Moreover, the attributes of the carbon source and its roughened structure, which yields a high surface area, may further promote nitrate removal capacity by facilitating microbial growth within the biofilm architecture, a phenomenon previously documented in analogous investigations [31].

The analysis of variance conducted indicated no significant differences between the behavior of nitrate concentration over time and the various initial concentrations at which the process was performed (*p*-value = 0.6318). The mean concentrations during the batch experiment were 34.77, 22.41, 9.86, and 8.13 mg/L for 115, 70, 30, and 20 mg/L, respectively. Furthermore, the reaction rate constants were determined to be $-0.198\ \mathrm{d^{-1}}$, $-0.211\ \mathrm{d^{-1}}$, $-0.197\ \mathrm{d^{-1}}$, and $-0.342\ \mathrm{d^{-1}}$, with $R^2$ values of 0.9681, 0.9405, 0.7009, and 0.9216 at initial concentrations of 20, 30, 70, and 115 mg/L, respectively. The reaction rate constant increased as the initial concentration increased. However, this trend was not observed when the initial concentration was 70 mg/L, where the reaction rate was lower ($-0.197\ \mathrm{d^{-1}}$) with an $R^2$ value of 0.7009. Additionally, this behavior corresponds to pseudo-first-order kinetics, implying that nitrate concentration directly influences denitrification. The half-life of the nitrate concentration reduction was 3.49, 3.29, 3.52, and 2.03 days for concentrations of 20, 30, 70, and 115 mg/L, respectively. The half-life was greater when the initial nitrate concentration was 70 mg/L. However, it decreased as the nitrate concentration increased. High initial nitrate concentrations resulted in a faster rate of decrease, consistent with the previously reported findings [20].

- Formation of nitrite and ammonium

Nitrite concentrations initially increased and then exhibited a tendency to decrease. This behavior aligns with previous research where nitrite concentration initially rose before declining, attributed to the biological denitrification process, wherein nitrate was consumed and converted into nitrite, subsequently transforming into nitrogen gas [20,59]. On the other hand, ammonium concentrations progressively decreased as denitrification occurred. Nitrite and ammonium concentrations were higher at a higher initial nitrate concentration of 115 mg/L, maintaining ranges of 0.01 to 0.05 mg/L and 0.05 to 0.26 ± 0.11 mg/L, respectively. According to the analysis of variance, there were no significant differences

in nitrite and ammonium concentrations over time when concentrations were different (*p*-value = 0.6861) and (*p*-value = 0.3835) for nitrite and ammonium, respectively.

The mean nitrite concentrations were 0.03, 0.01, 0.02, and 0.02 mg/L for initial nitrate concentrations of 115, 70, 30, and 20 mg/L, respectively. On the other hand, ammonium concentrations were 0.18, 0.08, 0.08, and 0.11 mg/L for initial concentrations of 115, 70, 30, and 20 mg/L, respectively. As reported in the literature, the consistently low levels of nitrate and ammonium concentrations suggest that nitrate removal predominantly occurred via denitrification, wherein nitrate was effectively reduced to nitrogen [30,51].

Furthermore, a moderately negative correlation was observed between nitrite and nitrate concentrations, with values of −0.35, −0.55, −0.8, and −0.58 for the initial nitrate concentrations of 20, 30, 70, and 115 mg/L, respectively. This indicates that nitrite concentration increased as nitrate concentration decreased, indicative of the denitrification process, as nitrite serves as an intermediate. Denitrification, a biological process wherein nitrate is sequentially reduced to nitrite and gaseous nitrogen oxides, likely contributed to the observed decrease in nitrate and increase in nitrite, consistent with the findings reported by Zhong et al. [60].

During the batch test, ammonium concentrations remained below 0.3 mg/L, suggesting that the corn cob matrix provided the necessary organic carbon to prevent $NH_4^+$-N presence in the effluent, maintaining concentrations below 1 mg/L, as noted in prior studies indicating low concentrations [20]. This low level of ammonium suggests that nitrate reduction predominantly occurred through denitrification rather than dissimilatory nitrate reduction to ammonium (DNRA), which is in line with previous research [17,35].

The positive correlation between ammonium and nitrate concentrations across all conditions, ranging from moderate to strong (0.98 to 0.6), indicates that as nitrate concentration decreases, so does ammonium concentration. Specifically, correlation coefficients of 0.98, 0.88, 0.92, and 0.60 were observed when initial nitrate concentrations were 20, 30, 70, and 115 mg/L, respectively. Providing an organic carbon-source-facilitated denitrifying microorganism activity by supplying the necessary energy and carbon for nitrate reduction reactions [4].

- Features of carbon release

The COD tended to increase when the initial concentrations were 115 mg/L and 30 mg/L, as shown in Figure 5, while an increase followed by a slight decrease was obtained when the initial concentrations were 20 and 70 mg/L, which coincides with what was obtained by Guan et al. [42]. The increase in COD indicated the release of carbon in denitrification, where abundant soluble and small-molecular-weight organic compounds were released; the nitrate removal performance was influenced by the number of materials released, which coincides with the results obtained by Li et al. [34]. However, there is no significant difference between the behavior of the chemical oxygen demand concerning the initial concentrations evaluated (*p*-value = 0.98).

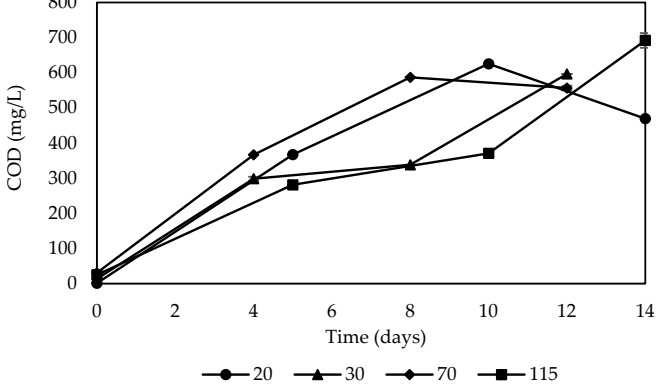

**Figure 5.** Carbon release behavior of corn cob during denitrification at different initial nitrate concentrations.

The chemical oxygen demand behaved similarly for the different concentrations; the COD averages were 385.13, 365.75, 342.19, and 311.63 mg/L for the initial concentrations of 70, 20, 115, and 30 mg/L, respectively. The COD means remained in a similar range, which could indicate that when nitrate concentrations in the medium are higher, the release of carbon through the corn cob provided a carbon source and support for the denitrifying bacteria necessary to perform the denitrification process, which has been proven by previous studies [31,61].

Due to its degradable hemicellulose and cellulose composition and the rough surface, the corn cob provided nitrate removal capacity. In this sense, it has been reported that substrates with a surface that is too smooth could cause problems in binding hydrolysis and denitrification bacteria, leading to unsatisfactory denitrification performance [31,34]. Therefore, the release of carbon at different nitrate concentrations is related to the characteristics of the corn cob as a carbon source, which coincides with the results obtained by Ling et al. [31]. However, the denitrification performance of corn cob carbon sources may deteriorate after a long operation [34].

The correlation between the behavior of the nitrate concentration and the chemical oxygen demand (COD) was found to have strong negative correlations, $-0.96$, $-0.89$, $-0.95$, and $-0.83$, for concentrations of 20, 30, 70, and 115 mg/L, respectively. The chemical oxygen demand increased as the nitrate concentration decreased due to the release of carbon in the denitrification process, which reduced nitrate in the water by denitrifying bacteria. In the same way, there is a moderate negative correlation between COD and ammonium for all conditions; the relationship coefficient was $-0.97$, $-0.61$, $-0.76$, $-0.89$, for concentrations of 20, 30, 70, and 115 mg/L, respectively. This correlation was stronger when the initial nitrate concentrations were 20 mg/L and 115 mg/L, and moderate when the initial concentrations were 30 and 70 mg/L; as the COD concentration increased, the ammonium concentration decreased. As stated by Li et al. [34], the greater removal of nitrates could be due to the greater degradability of the corn cob due to the gradual maturation of the biofilms and the stable use of carbon sources during the stabilization of denitrification after the acclimatization of the denitrifying bacteria. On the other hand, a rapid decrease in COD after a brief initial period of increase may be due to the adhered biofilm gradually maturing in a short time [42]. Low carbon release can lead to slow biofilm growth, resulting in low nitrogen removal efficiencies and longer onset times [31].

- Environmental conditions during nitrate removal

During denitrification, dissolved oxygen was maintained below 3 mg/L, generating favorable anoxic conditions for denitrifying bacteria [62]. The variance analysis indicated no significant differences between oxygen concentrations and variations in initial nitrate concentration ($p$-value = 0.9475). The dissolved oxygen means were maintained at 2.97, 2.63, 2.06, and 1.79 mg/L for 20, 115, 30, and 70 mg/L concentrations, respectively. The Pearson correlation coefficient was 0.93, 0.99, 1, and 0.99 for concentrations of 20, 30, 70, and 115 mg/L, respectively, indicating that the DO was lower as the nitrate concentration decreased.

The pH was maintained between 5.5 and 7, ideal conditions for denitrifying bacteria and nitrate removal [63]. These values are consistent with those reported by Xie et al. [30]. pH plays a crucial role in denitrification, influencing the growth and metabolism of microorganisms, and enzymatic activity [62,64].

The variance analysis indicated no significant differences between the behavior of pH and the different initial concentrations ($p$-value = 0.6985). The mean pH values were 6.57, 6.56, 6.44, and 6.31 when the initial concentrations were 115, 70, 30, and 20 mg/L, which indicated that the pH was higher at higher nitrate concentrations. The correlation analysis showed a coefficient of 0.51 to 0.86 for concentrations of 20 mg/L and 30 mg/L, which could indicate that when pH conditions decrease, it decreases nitrate. The ratings were $-0.95$ and $-0.96$ for the highest concentrations of 70 and 115 mg/L, which could indicate that the nitrate concentration decreases while the pH conditions increase, which could be due to higher initial concentrations to which the denitrification process was carried out. On the other hand, the increase in nitrite concentration and the decrease in

pH value could indicate the denitrification effect [30]. The increase in pH was related to alkalinity production because of denitrification [18]. However, it can decrease when an acidic hydrolysate is released as a product of microorganisms' decomposition of carbon sources [62]. In a study, the total nitrogen removal rate experienced a trend of initially increasing and then decreasing as the pH decreased [65].

On the other hand, for the electrical conductivity concentrations, the statistical analysis indicated that there are significant differences between the electrical conductivity and the initial concentrations of nitrate (*p*-value = 0.0001), so the electrical conductivity was greater at higher initial concentrations of nitrate. This could be due to a greater presence of ions associated with higher levels of nitrate in water [66]. On the other hand, lower nitrate concentrations and, therefore, electrical conductivity could favor the activities of denitrifying microorganisms for greater nitrate removal efficiency [67].

### 3.2.3. Column Test

A column test was carried out to evaluate nitrate removal at different hydraulic retention times of 7, 24, and 16 h with an average initial concentration of $30.88 \pm 1.50$ mg/L of $NO_3^-$-N and an average ambient temperature of 28 °C. In this sense, the denitrification performance, nitrite and ammonium formation, and carbon release were analyzed and evaluated using the different HRTs.

- Denitrification performance

The effect of HRT on the denitrification rate and nitrate removal efficiency was studied, as shown in Figure 6. The denitrification rate was greater as the HRT was shorter; at the end of the experiment, the denitrification rates were 83.53, 43.00, and 32.82 mg $NO_3^-$-N/L for the 7, 16, and 24 h HRTs, respectively.

Therefore, a shorter HRT and higher flow rate increased the denitrification rate in the corn cob column system, which can be associated with a higher amount of nitrate in the water within a day. These results are consistent with the results reported by Xu et al. [21].

On the other hand, removal efficiencies were 74.09%, 87.13%, and 99.86% for the hydraulic retention times (HRTs) of 7, 16, and 24 h, respectively. A steady state of removal efficiency greater than 99% was observed at an HRT of 24 h, while slight fluctuations were noted at 16 and 7 h, followed by a decrease. These results align with previous research using corn cob as a reactive material, where nitrate removal efficiency diminished with shorter HRTs [20,21]. This is attributed to the prolonged interaction between corn cobs and microorganisms, facilitating denitrification, while a short HRT lead to faster water flow, cleaning, and separating microorganisms and solubilized substrates [68].

The groundwater velocities previously reported in the literature are slow [13,17,69]. Therefore, a hydraulic retention time (HRT) of 24 h could simulate these groundwater flows, suggesting that the application of permeable reactive barriers using corn cobs could be effective in removing nitrate concentrations in groundwater under humid tropical conditions. In this study, no additional denitrifying bacteria were added; the corn cobs provided the necessary indigenous denitrifying bacteria for nitrate removal at various concentrations, as studied in the batch test, using potassium nitrate fertilizer, which is commonly used in agricultural activity. Additionally, these results indicated that even with a faster flow, such as that used with a 7 h HRT, nitrate removal was also observed, suggesting that nitrate removal under these conditions and using this reactive material can occur under a variety of water flow conditions, depending on soil characteristics. Therefore, the use of corn cobs as a reactive material in permeable reactive barriers with a 24 h HRT could be a nature-based solution to this global issue.

The analysis of first-order kinetics applied to nitrate concentrations across various HRTs indicated a better fit with pseudo-first-order kinetics. The coefficients of determination ($R^2$) were 0.472, 0.7799, and 0.9304, and the reaction rate constants (k) were 0.094 $d^{-1}$, 0.142 $d^{-1}$, and 0.539 $d^{-1}$ for HRTs of 7, 16, and 24 h, respectively. These results demonstrate that increasing HRT enhances the reaction rate, thereby improving nitrate removal efficiency. For instance, at an initial nitrate concentration of 30.88 mg/L and an HRT of 7 h,

the concentration halved in 7.40 days, whereas at an HRT of 24 h, it took only 1.29 days. These findings align with those reported by Liu et al. [20], who observed that reaction rate constants increased with longer HRTs.

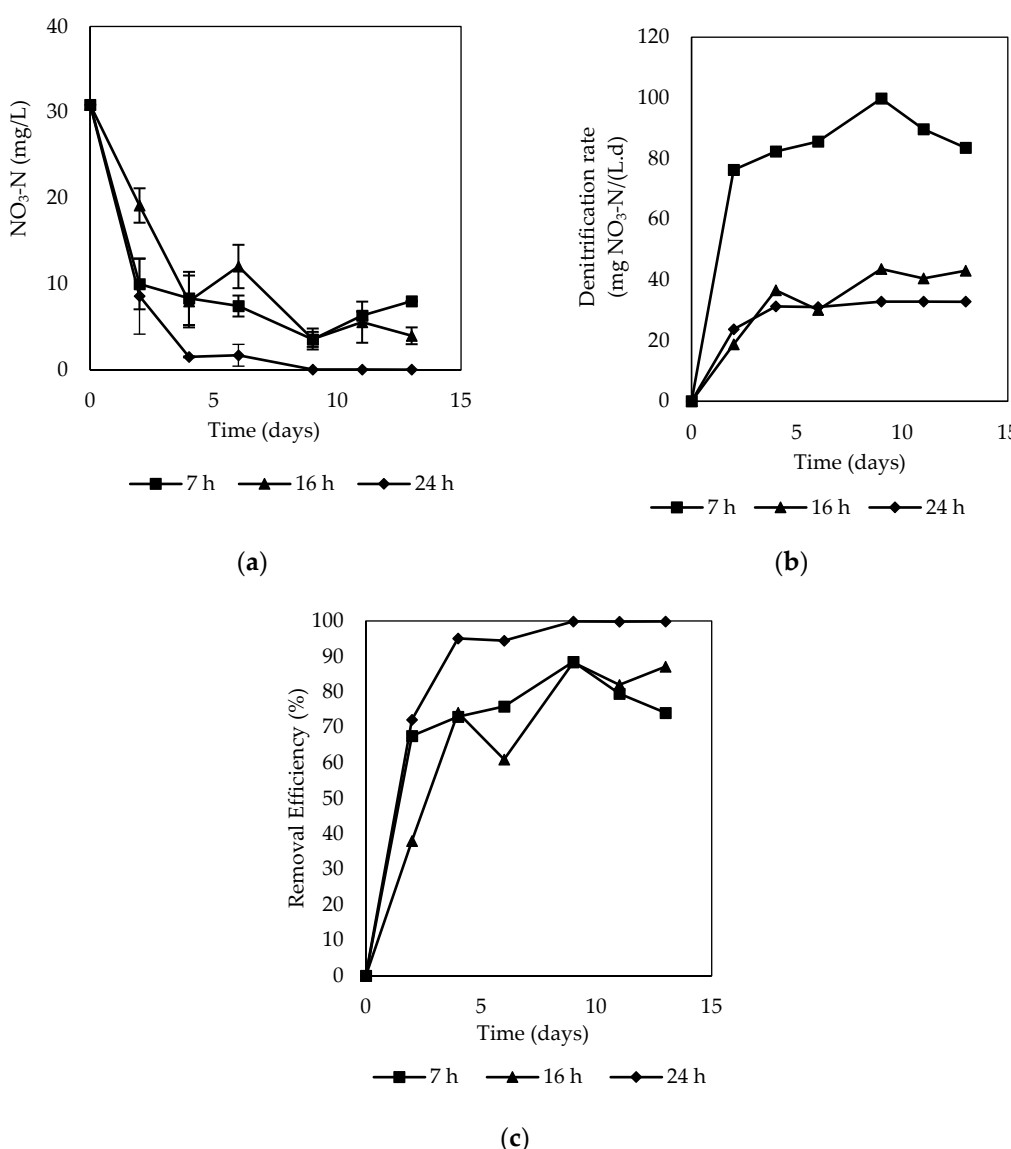

**Figure 6.** Performance of denitrification using corn cob: (**a**) behavior of nitrate concentration at various HRTs; (**b**) nitrate removal efficiency; (**c**) denitrification rate.

- Formation of nitrite and ammonium during denitrification

For the 7 h and 24 h HRTs, nitrite concentrations remained below 0.2 mg/L. Specifically, during the 7 h HRT, nitrite concentrations decreased and stayed within the range of 0.08 to 0.18 mg/L and an average of 0.12 ± 0.04 mg/L. For the 24 h HRT, the concentration was even lower, with an average of 0.06 ± 0.07 mg/L, and fluctuated between 0.02 to 0.2 mg/L. However, when the HRT was reduced to 16 h, nitrite concentration spiked to a maximum of 1.79 mg/L, which is significantly higher than the levels observed at 7 and 24 h. The average concentration at this HRT was 0.77 mg/L, ranging from 0.03 to 1.79 mg/L. This increase in nitrite accumulation at shorter HRTs could be attributed to diminished physical contact between the solution and denitrifying bacteria, as noted in [68].

Ammonium concentrations were generally maintained below 3.63 mg/L. Using a 7 h HRT, concentrations ranged from 0.10 mg/L to 2.25 mg/L, with an average of

$0.87 \pm 0.79$ mg/L. The highest ammonium concentrations were recorded during the 24 h HRT, averaging $1.85 \pm 1.52$ mg/L. Initially, ammonium concentrations exhibited a decreasing trend, which was followed by significant increases, peaking at 3.63 mg/L. Conversely, with a reduced HRT of 16 h, the highest ammonium concentration was 1.38 mg/L, with a minimum of 0.53 mg/L and an average of $1.04 \pm 0.33$ mg/L. Ammonium concentrations were higher with longer HRTs. However, statistical analysis revealed no significant differences between the ammonium concentrations and the HRTs evaluated ($p$-value = 0.2298).

Ammonium concentrations were higher with a 24 h HRT, while nitrite concentrations were lower. In contrast, using a 16 h HRT, ammonium concentrations were lower, but the highest nitrite concentrations were observed. On the other hand, nitrite and ammonium concentrations using a 7 h HRT remained within an intermediate range. Despite these variations, as reported in the literature, the effluent ammonium concentration remained consistently low throughout the experiment [68].

- Characteristics of carbon release

The COD decreased over time, as illustrated in Figure 7. Initially, there was an increase at the start of the experiment, with a COD of 1838.50 mg/L for a 7 h HRT, which then gradually declined. The average concentrations were 823.18 mg/L, 276.50 mg/L, and 92.46 mg/L, respectively, corresponding to the HRTs of 7, 24, and 16 h. This pattern aligns with the sequence in which the HRTs were evaluated during the operation time of the columns, indicating that COD decreased as the experiment progressed. These results are consistent with findings reported by other researchers, where carbon release was greater at the beginning of the experiment and decreased over time [21,70,71]. This increase in COD is attributed to the soluble organic carbon released through autolysis, which is rapidly consumed by microorganisms. Water-soluble substances from the corn cob served as a carbon source, gradually dissolved in the water and eliminated in the effluent over time. As the experiment progressed, the number of microorganisms increased, using a more soluble fraction of carbon, resulting in a decreased relative percentage of soluble components.

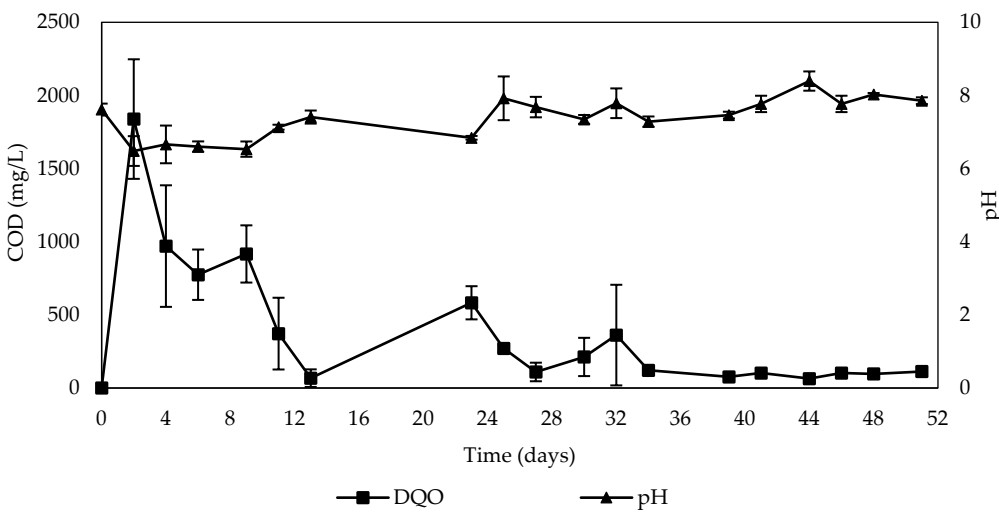

**Figure 7.** Behavior of carbon release and pH during nitrate removal in the column test.

The steady state of carbon release may be attributed to the slowed release rate of the carbon source, resulting from the inhibition of corn cob decomposition by its lignin content, as reported by other authors [21,71]. Carbon release from solid carbon sources can be divided into two stages. The first stage involves rapid release due to the liberation of easily decomposable carbohydrates and the swift degradation of water-soluble substances. Small carbon molecules adhere to the material's surface and then separate rapidly due to their expansion and dissolution in the aqueous solution. The second stage features a slower release, where carbon release diminishes until an equilibrium state is reached.

Consequently, the decomposition rate decreases as small carbon-containing molecules within the material are broken down, dissolved, and released into the water. The easy degradation of the material is limited by the accumulation of substances such as carbon dioxide and the presence of difficult-to-degrade components like lignin [42].

On the other hand, the pH exhibited fluctuations with a tendency to increase, as shown in Figure 7. Initially, the pH ranged from 6.48 to 7.41 during the 7 h HRT. This subsequently increased, stabilizing between 6.84 and 7.92 for the 24 h HRT. The maximum pH was observed during the 16 h HRT, remaining between 7.46 and 8.39. The average pH values were $6.80 \pm 0.38$, $7.48 \pm 0.4$, and $7.88 \pm 0.31$, corresponding to the HRTs of 7 h, 24 h, and 16 h, respectively. Therefore, the pH increases as the operation time and the denitrification process progress. This rise is associated with the decrease in COD, indicating a strong relationship between pH and COD relative to the operation time, which aligns with the sequence in which HRT influences these variables (7 h, 24 h, and 16 h).

Significant differences were found between the pH levels and the evaluated HRTs ($p$-value = 0.0005). However, no significant differences were observed between the 24 h and 16 h HRTs, although both significantly differed from the pH during the 7 h HRT, according to Tukey's analysis ($p$-value < 0.05). The increase in pH may be due to alkalinity production during heterotrophic denitrification, as nitrite is reduced to nitrogen gas [68]. An optimal pH for the denitrification process ranges between 5.5 and 8.0, which favors the growth and activity of denitrifying bacteria populations, thereby enhancing nitrate removal. This is because pH significantly affects microbial activity, influencing denitrifying communities' size and composition [33].

Dissolved oxygen levels were maintained between 2.10 mg/L and 7.54 mg/L, with an average concentration of $5.35 \pm 1.5$ mg/L throughout the experiment. The average dissolved oxygen concentrations were $6.30 \pm 0.73$ mg/L, $5.70 \pm 1.09$ mg/L, and $3.81 \pm 1.2$ mg/L, corresponding to the HRTs of 16 h, 7 h, and 24 h, respectively. Significant differences were observed between the dissolved oxygen levels and the evaluated HRTs ($p$-value = 0.0016). However, the dissolved oxygen levels for the 7 h and 16 h HRTs were not significantly different, while they were significantly different from the 24 h HRT, according to Tukey's analysis ($p$-value > 0.05).

The dissolved oxygen was lower with a 24 h HRT, reaching a minimum concentration of 2.10 mg/L. This lower level could be inferred to provide a suitable environment for denitrifying bacteria, thereby facilitating a decrease in nitrate concentration and enhancing removal efficiency. Denitrification is anaerobic; thus, a low dissolved oxygen concentration could promote greater nitrate removal efficiency. However, even when the dissolved oxygen levels were above 4 mg/L, the removal efficiency still exceeded 50%. These results are consistent with findings reported by Xu et al. [21].

The Pearson correlation analysis was conducted to understand the relationship between the variables. Statistically significant relationships were found ($p$-value < 0.05) in two groups: one involving the concentrations of nitrate, nitrite, and dissolved oxygen in the effluent and the other involving COD, pH, and electrical conductivity. The Pearson correlation showed a coefficient of 0.76 between the concentrations of nitrate and nitrite, suggesting a moderately strong tendency for these concentrations to increase together in a linear relationship, which is statistically significant ($p$-value = 0.0003). The relationships between nitrate and nitrite with dissolved oxygen were also statistically significant, with Pearson correlation coefficients of 0.51 and 0.50 and $p$-values of 0.0295 and 0.0356, respectively. This indicates an average tendency for one variable to decrease as the other decreases, confirming the significance of these relationships.

COD and pH demonstrated a Pearson correlation of $-0.79$, indicating an inverse relationship where the one variable tends to decrease as the other variable increases. This aligns with previous observations that pH tends to increase while COD decreases over operation time. This relationship is statistically significant, with a $p$-value of 0.0001. On the other hand, COD and electrical conductivity exhibited a Pearson correlation of 0.75 with a $p$-value of 0.0004, suggesting a highly positive and statistically significant correlation.

As the COD increases, electrical conductivity also tends to rise, and vice versa. Electrical conductivity showed higher concentrations at the beginning of the experiment, followed by a decreasing trend. The average electrical conductivity readings were 1040.15 μS/cm, 997.47 μS/cm, and 975.40 μS/cm for HRTs of 7 h, 16 h, and 24 h, respectively, indicating that electrical conductivity was greater when the HRT was shorter. However, no significant differences were found between electrical conductivity and the influence of HRT, with a *p*-value of 0.3613.

### 3.3. Surface Characteristics of the Carbon Source

The surface characteristics of the corn cob were obtained by scanning electron microscopy (SEM), spectroscopy, and Fourier transform infrared, so the corn cobs used during batch testing and column testing were analyzed before and after the denitrification treatment.

### 3.3.1. Scanning Electron Microscopy

Figure 8 shows the distribution of SEM images of the corn cob before and after carbon release at different concentrations in the batch test, and displays images of corn cobs at various initial nitrate concentrations used during the batch test: 20, 30, 70, and 115 mg/L. Thus, "M" represents the fresh corn cob before denitrification, while M20, M30, M70, and M115 represent the corn cob after denitrification at these different concentrations, respectively. Figure 9 presents images of the corn cob post-treatment in the column test, which lasted 52 days at different hydraulic retention times (HRTs) with an initial concentration of 30.88 ± 1.50 mg/L. M1, M2, and M3 denote the corn cobs derived from columns 1, 2, and 3, respectively.

The morphology of the surface of the carbon source used during nitrate removal plays a crucial role as it significantly influences the growth and reproduction of microorganisms. Therefore, observing changes in the microstructure of agricultural waste, specifically corn cob, before and after treatment helps to assess its feasibility as a carbon source and a carrier for microorganisms. This study noted that the corn cobs exposed to denitrification treatment exhibited more cracks and irregularities compared to the fresh corn cobs, as depicted in Figures 8 and 9. The treated corn cobs showed increased roughness, with more convex and porous structures, thereby enhancing their performance as a support for denitrifying bacteria due to improved adhesion and growth of microbial communities on the surface of the corn cob. This observation aligns with the findings from other studies that used agricultural waste such as cane straw and corn cob as carbon sources [34,53,72]

Additionally, it was observed that the corn cob maintained its stable physical structure after the treatment, as no significant detachments were noted. This stability suggests that the corn cob possesses a robust structure, which could be attributed to the composition of the lignin contained within it. Lignin is known for its resistance to degradation, reinforcing the structural integrity of the corn cob. Therefore, agricultural waste can serve as a carrier of biofilms. This phenomenon has been reported by Feng et al. [35] and Yang et al. [53]. Furthermore, in Figure 9h, a surface composed of hairs and small holes was observed, consistent with the results Yang et al. reported [53].

Figure 10 shows the pith part of the corn cob before and after the release of carbon in the column test. It was observed that the pore structure and size increased after carbon release. In Figure 10a,b, a homogeneous structure with pores was observed in the fresh cob; the pith is similar to a sponge with thin walls that contain small holes and cellulose microfibrils on the surface of the cell bodies. In addition, it has good water capacity and a high specific surface area for cellulose, which allows for the contact between cellulose and microorganisms to accelerate the decomposition process, which has been previously reported by other authors. In Figure 10c,d, it was observed that in the cob, the size of the pores increased after denitrification, and a more significant decomposition of the structure was observed. The growth of pores allowed for a greater space for the union and the growth of denitrifying bacteria. This phenomenon was also evident in the results obtained by Yang et al. [53].

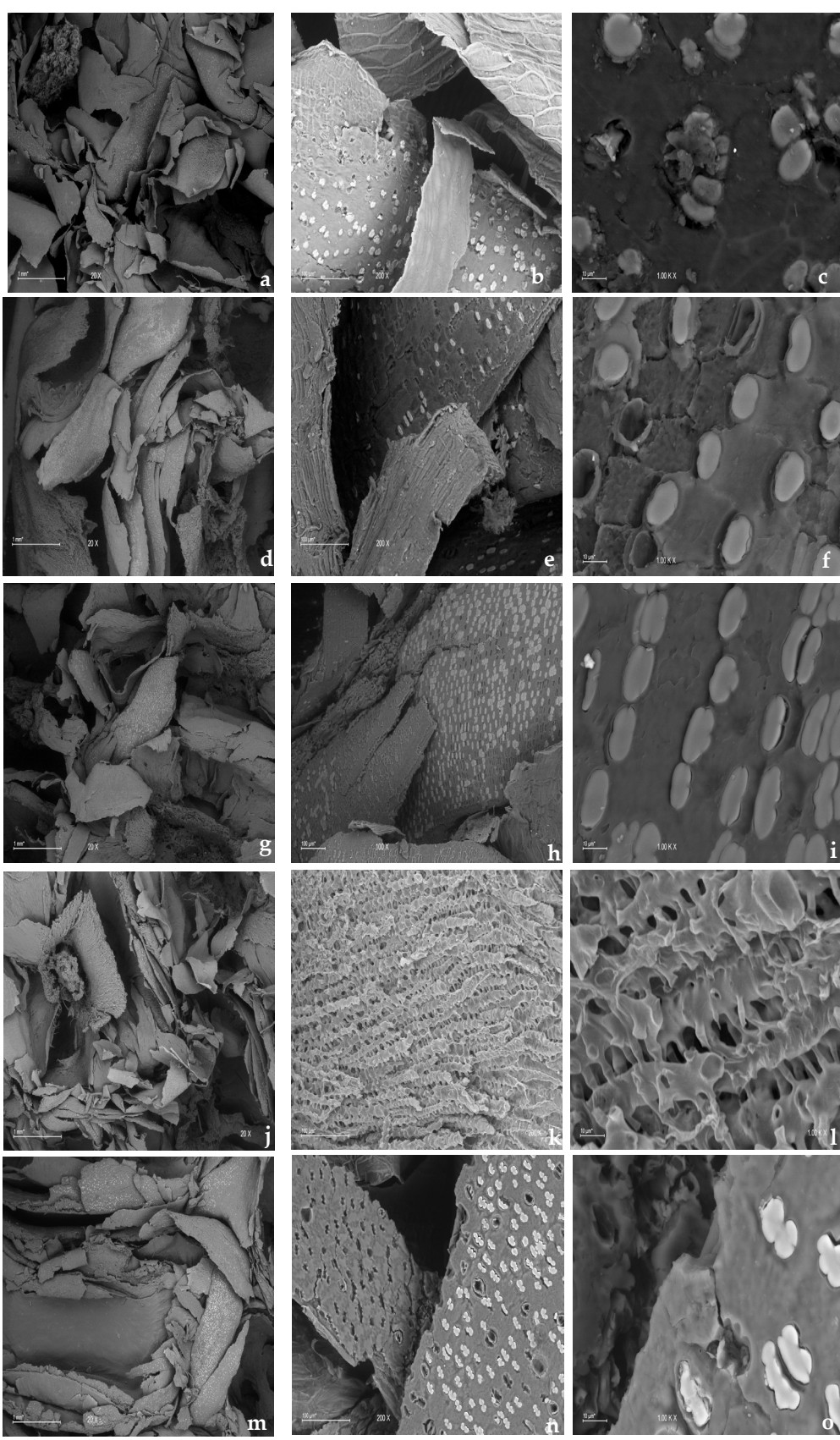

**Figure 8.** Distribution of SEM images of the corn cob before and after carbon release at different concentrations in the batch test. (**a**) Fresh corn cob before denitrification taken at 20×, 1 mm; (**b**) Fresh

corn cob before denitrification taken at 200×, 100 μm; (**c**) Fresh corn cob before denitrification taken at 1 K×, 10 μm; (**d**) Corn cob after denitrification at a concentration of 20 mg/L taken at 20×, 1 mm; (**e**) Corn cob after denitrification at a concentration of 20 mg/L at 200×, 100 μm; (**f**) Corn cob after denitrification at a concentration of 20 mg/L taken at 1 K×, 10 μm; (**g**) Corn cob after denitrification at a concentration of 30 mg/L taken at 20×, 1 mm; (**h**) Corn cob after denitrification taken at a concentration of 30 mg/L at 200×, 100 μm; (**i**) Corn cob after denitrification at a concentration of 30 mg/L taken at 1 K×, 10 μm; (**j**) Corn cob after denitrification at a concentration of 70 mg/L taken at 20×, 1 mm; (**k**) Corn cob after denitrification at a concentration of 70 mg/L taken at 200×, 100 μm; (**l**) Corn cob after denitrification at a concentration of 70 mg/L taken at 1 K×, 10 μm; (**m**) Corn cob after denitrification at a concentration of 115 mg/L taken at 20×, 1 mm; (**n**) Corn cob after denitrification at a concentration of 115 mg/L taken at 200×, 100 μm; (**o**) Corn cob after denitrification at a concentration of 115 mg/L taken at 1 K×, 10 μm.

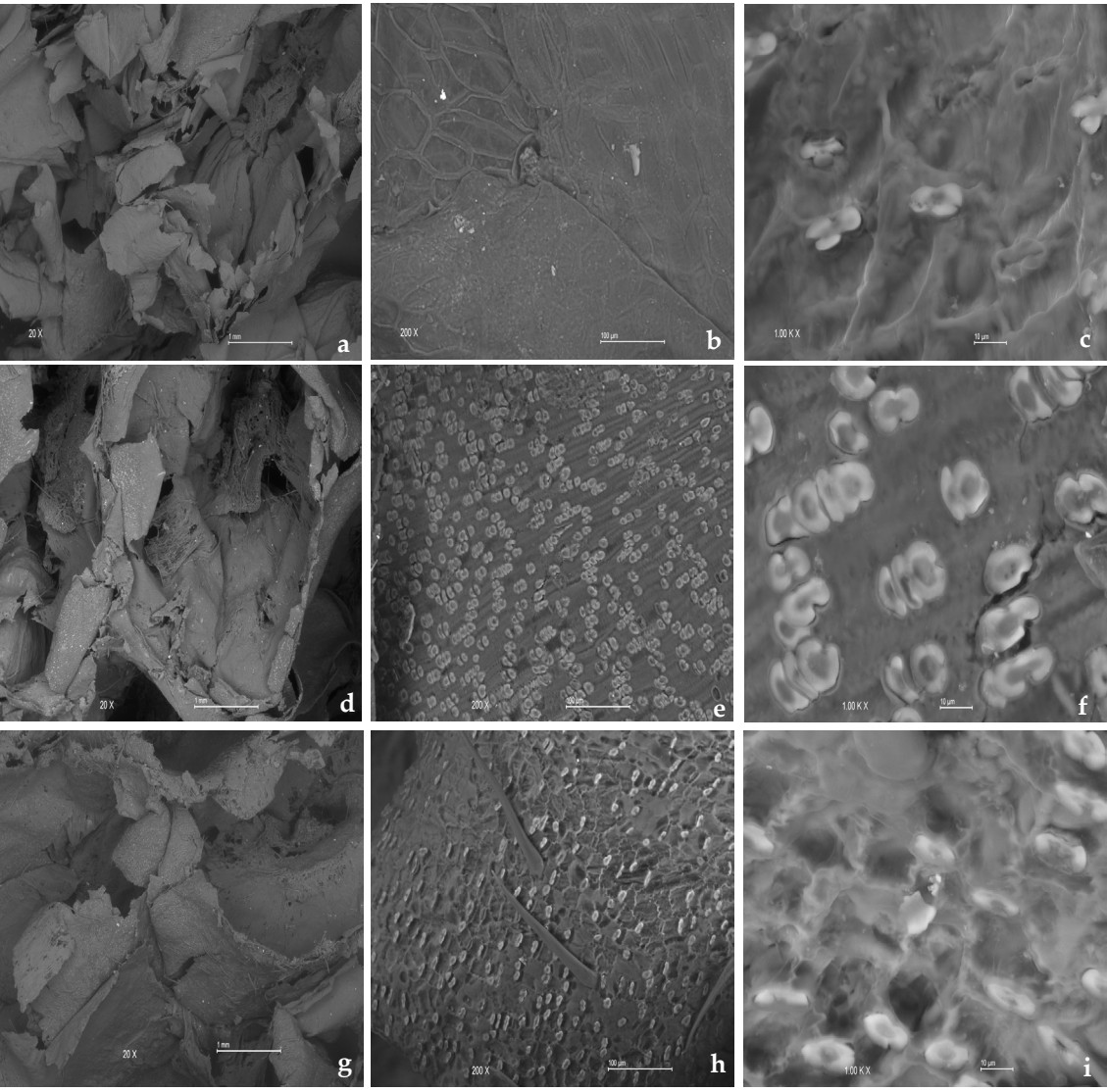

**Figure 9.** Distribution of SEM images of corn cob after carbon release in the column test. (**a**) Corn cob from column 1 taken at 20×, 1 mm; (**b**) corn cob from column 1 taken at 200×, 100 μm; (**c**) Corn cob from column 1 taken at 1 K×, 10 μm; (**d**) Corn cob from column 2 taken at 20×, 1 mm; (**e**) Corn cob from column 2 taken at 200×, 100 μm; (**f**) Corn cob from column 2 taken at 1 K×, 10 μm; (**g**) Corn cob from column 3 taken at 20×, 1 mm; (**h**) Corn cob from column 3 taken at 200×, 100 μm; (**i**) Corn cob from column 3 taken at 1 K×, 10 μm.

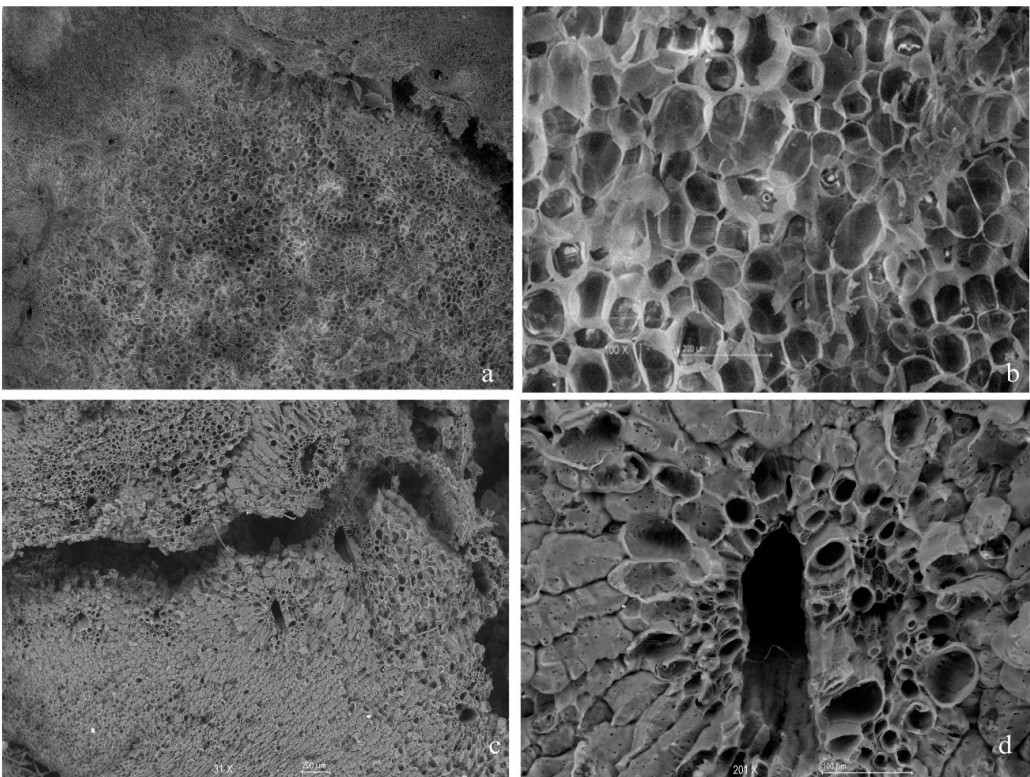

**Figure 10.** Distribution of SEM images of corn cob pith before and after carbon release in the column test: (**a**) fresh corn cob taken at 16×, 1 mm; (**b**) fresh corn cob taken at 100×, 200 μm; (**c**) used cob taken at 31×, 200 μm; (**d**) used cob taken at 201×, 100 μm.

The elemental carbon on the surface of the carbon source was analyzed by energy dispersive X-ray spectroscopy (EDS), and it was found that compared with the fresh carbon source, the distribution intensity of the carbon element in the carbon source after release increased to various degrees. The fresh corn cob represented $52.64 \pm 0.22\%$ of the total atomic distribution, while it increased to 60.21%, 55.84%, 59.83%, and 60.61% after carbon release in the batch test when an initial concentration of 20, 30, 70, and 115 mg/L, respectively. On the other hand, for the corn cobs used in the column test, the distribution intensity of the carbon element was 66.45%, 76.58%, and 74.96% for the corn cobs from column 1, column 2, and column 3, respectively. This showed an increase due to the release of carbon from the corn cob, allowing for microorganisms to use it as a carbon source for denitrification easily. Furthermore, the distribution of C was more significant in the corn cobs used in the column tests compared to those in the batch test. This could be attributed to a longer exposure time to the denitrification treatment, resulting in a greater carbon release.

Consequently, a higher distribution of carbon was found on the surface of the corn cobs after the denitrification treatment. These results are consistent with those reported by Feng et al. [35]. In contrast to what was reported by Li et al. [34], no microorganisms, such as cocci or rod-shaped microorganisms, were observed.

### 3.3.2. Spectroscopy and Fourier Transform Infrared (FTIR)

To evaluate the changes in the chemical structure of the carbon sources before and after the experiments, spectroscopy and Fourier transform infrared (FTIR) analysis was used. This approach helps analyze the release of carbon sources, which is tied to the intermolecular bonds. The fresh corn cob displayed a spectral appearance similar to the corn cobs used at different initial nitrate concentrations (Figure 11a). This similarity was also observed by Gan et al. [73]. However, a more pronounced difference was noted between the fresh cob and those after treatment, where samples of corn cob from each column were analyzed, as shown in Figure 11b.

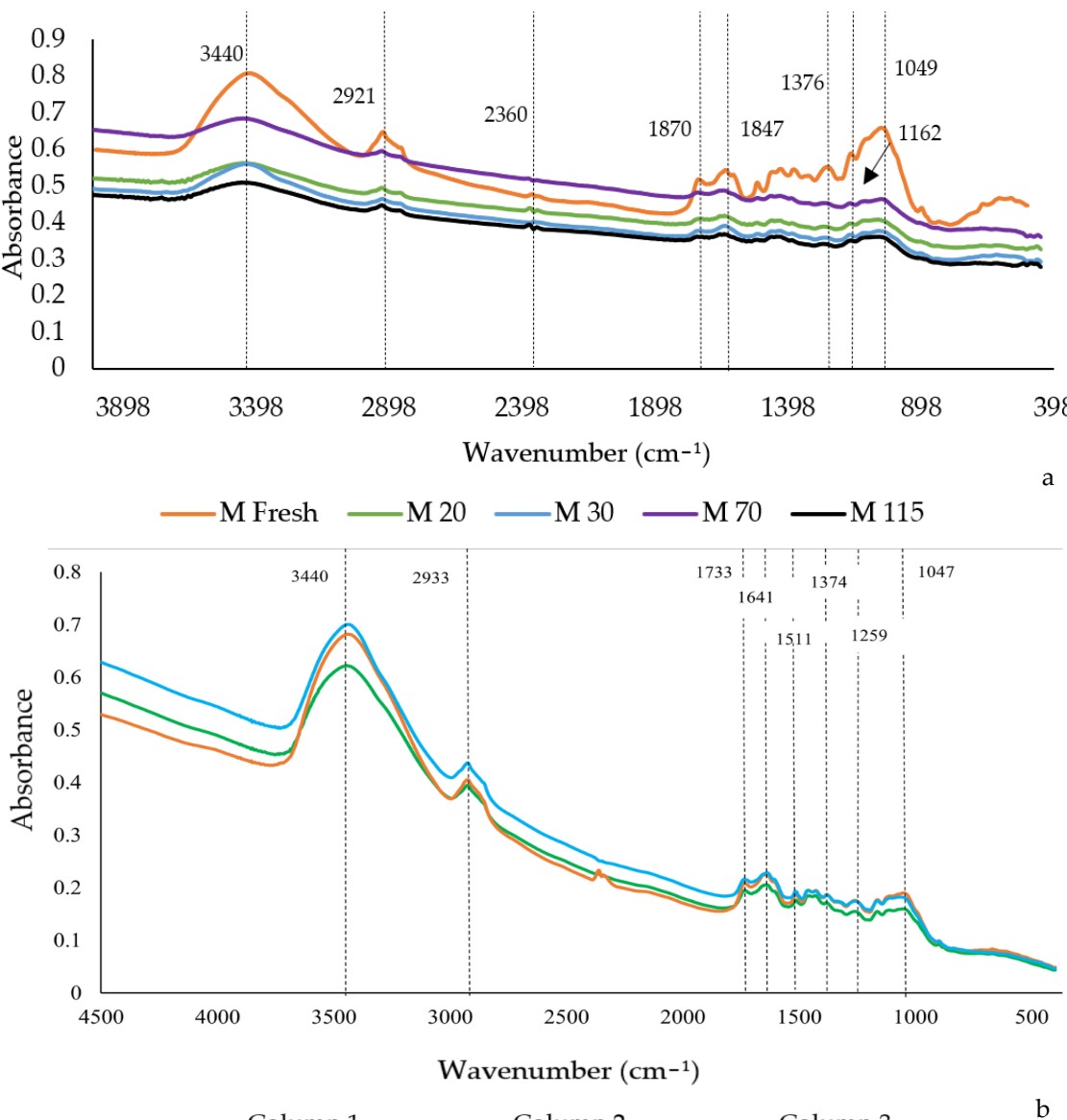

**Figure 11.** Comparison of FTIR spectra of fresh and used corn cob in the batch test and the column test. (**a**) Behavior of the FTIR spectra at different initial nitrate concentrations in the batch test; (**b**) behavior of the FTIR spectra at different HRTs in the column test.

The fresh cob (M Fresh) sample exhibited more pronounced peaks compared to the samples from Cob 20 (M20), Cob 30 (M30), Cob 70 (M70), Cob 115 (M115) (Figure 11a), and the column samples (Figure 11b). The absorption peaks near 896 cm$^{-1}$ observed in the fresh cob and at different concentrations are attributed to the β-glucoside bonds, corresponding to cellulose's C-H bending vibration functional group [40]. However, this wavenumber was not prominently observed in the samples from the column tests (Figure 11b). The presence of cellulose in the biomass is reduced by cellulose hydrolysis [74], which might explain the decreased presence in the column tests where the denitrification process was conducted over a more extended period, potentially leading to more extensive biomass degradation (Figure 11b). The diminished peak at 869 cm$^{-1}$ after treatment suggests that many of the ring structures in the column samples were disrupted during the experiment [35]. Additionally, a peak close to 871 cm$^{-1}$ could indicate the presence of polysaccharides [75].

Bands below 1000 cm$^{-1}$ in the "fingerprint region" are associated with unsaturated bonds [72,76]. The wavenumbers close to 1049 cm$^{-1}$ and 1047 cm$^{-1}$, observed in the fresh corn cob and the samples at varying nitrate concentrations and in the column test

samples, respectively, are attributed to the stretching vibration functional group of C-O bonds in cellulose. The peaks around 1027 cm$^{-1}$ are generated by the bending movements of C-C, C-O, and C-OH bonds, indicating a significant increase in polysaccharides [75]. The absorption peaks of fresh corn cob after treatment of 1376 cm$^{-1}$ (Figure 11a) and 1374 cm$^{-1}$ (Figure 11b) were attributed to the bending vibrations of CH$_3$ [34]. While the absorption peak close to 1376 cm$^{-1}$ (Figure 11a,b) corresponds to the vibration of the C-H bond [40]. A characteristic of biomasses that is composed of cellulose and hemicellulose.

The vibrations close to 1420 cm$^{-1}$ represent the C-C functional groups present in lignin. The C-O stretch bond associated with lignin is found near 1259 cm$^{-1}$. Additionally, the absorption peak close to 1511 cm$^{-1}$ corresponds to the aromatic bond C=O, representing the primary substance of lignin [40]. The absorption peak of 1511 cm$^{-1}$ (Figure 11b) could be attributed to amide I and II bands of cellular proteins. Authors have suggested that the effluent may contain proteins and soluble microbial byproducts, which are organic compounds derived from microbial metabolites [34]. Nitrate absorptions can be observed in the peaks ranging from 1200 cm$^{-1}$ to 1500 cm$^{-1}$ in the cob samples at different concentrations [73] (Figure 11a) and the samples obtained from the column experiment (Figure 11b). However, the waves were difficult to observe because their intensity was lower than the intensity of the presence of water in the samples (in the range of 3000–3800 and 1500–1800 cm$^{-1}$).

The spectra of samples at different concentrations did not show differences in wave intensity (Figure 11a), nor did they display an absorption trend with increasing nitrate concentrations, contrary to what was proposed by Gan et al. [73]. This may be due to the short duration during which the samples were exposed to different concentrations (14 and 12 days) in the Batch assay. In the column samples, the intensity of the water absorption waves was higher than in the samples at different concentrations, which could be attributed to the more prolonged contact with water during the column test (52 days) (Figure 11b). In Figure 11b, a peak at 1641 cm$^{-1}$ can be observed, which can be assigned to the bending of CO groups in proteins [75]. The absorption peak at 1733 cm$^{-1}$ belongs to the C=O stretching functional group [35], characteristic of aldehydes and carboxyl compounds typically found in biomasses with hemicellulose [40] and lignin [76] content. On the other hand, the peaks at 2921 cm$^{-1}$ (Figure 11a) and 2933 cm$^{-1}$ (Figure 11b) are in the C-H bond, common in cellulose. Near 3424 cm$^{-1}$ and 3440 cm$^{-1}$, the descending section in cellulose is identified [40]. Furthermore, these frequencies correspond to the vibration of functional groups O-H, N-H (stretching vibrations), and C-H, as found in amines and carboxylic acids, representing types of organic matter [72].

The cob and column samples exhibit a strong intensity of these frequencies. In the case of the column samples; this could be attributed to the binding of water to the surface of the biomass following the biological experiment [76], suggesting the presence of more hydrophilic functional groups. Conversely, the samples of cob 20, cob 30, cob 70, and cob 115 show weak peaks at 2921 cm$^{-1}$, which belong to the O-H group. This could indicate that these samples have a weaker water binding capacity than the column samples and the untreated cob [76].

## 4. Conclusions

Corn cob was evaluated as a carbon source for groundwater denitrification, specifically as a reactive material for use in permeable reactive barriers for groundwater remediation. Analysis revealed that corn cob from Panama primarily consists of hemicellulose, cellulose, and lignin at 50.8%, 34%, and 6%, respectively. During the leaching test, the corn cob released organic components and nitrogenous species at levels that pose no risk of secondary contamination.

The nitrate removal efficiency was recorded at 93.14% for an initial concentration of 22.18 ± 2.82 mg/L, 91.58% for 27.3 mg/L, 90.77% for 69.1 ± 1.2 mg/L, and 98.74% for 115.08 ± 1.88 mg/L. Thus, the removal efficiency generally decreased as the initial concentration increased. However, the efficiency remained above 90%, indicating that nitrate removal was predominantly achieved through denitrification. The release of carbon (COD) facilitated adequate biofilm growth, resulting in high removal efficiency.

HRT significantly affected nitrate removal, with greater efficiency observed at 24 h compared to the 7 h and 16 h conditions. Specifically, elimination efficiencies were 74%, 87.13%, and 99.86% for HRTs of 7 h, 16 h, and 24 h, respectively. Extending the HRT beyond 16 h could significantly enhance nitrate removal efficiency in groundwater at an initial concentration of 30.88 mg $NO_3^-$-N/L. The reaction rate constants were 0.094/day, 0.142/day, and 0.539/day for HRTs of 7, 16, and 24 h, respectively, indicating that increasing the HRT significantly accelerated the reaction rate, positively influencing nitrate removal. The denitrification rates were higher at shorter HRTs, calculated at 83.53, 43, and 32.82 mg $NO_3^-$-N/L/day for 7 h, 16 h, and 24 h, respectively.

The pH was maintained within the optimal ranges necessary for the denitrification process. Although dissolved oxygen levels exceeded 4 mg/L during the column test, the concentrations of ammonium and nitrite remained low, contributing to the effective elimination of nitrate. SEM images taken after the denitrification process showed no apparent fractures in the corn cob; instead, the surface appeared more porous, potentially enhancing microbial adhesion and facilitating denitrification. Additionally, the observed increase in carbon composition post-treatment indicated a release of carbon. Thus, corn cob could serve as a viable carbon source for denitrifying bacteria in the remediation of nitrate-contaminated groundwater, employing a permeable reactive barrier for in situ applications.

**Author Contributions:** Conceptualization, E.D. and G.C.S.H.; methodology, G.C.S.H., E.D. and M.D.L.Á.O.; data curation G.C.S.H.; writing: preparation of the original draft, G.C.S.H.; writing: review and editing, G.C.S.H., E.D. and M.D.L.Á.O.; financing acquisition, E.D. All authors have read and agreed to the published version of the manuscript.

**Funding:** This work was funded by the Secretaría Nacional de Ciencia, Tecnología e Innovación de Panamá (SENACYT) under the project number FIED19-R2-018, titled "Evaluación de Alternativas de Tratamientos Sostenibles para Remover Nitrato de Aguas Contaminadas", and Master of Science in Mechanical Engineering program, VI Cohort. Additional funding was provided by the Sistema Nacional de Investigación (SNI).

**Data Availability Statement:** The original contributions presented in the study are included in the article, further inquiries can be directed to the corresponding author.

**Acknowledgments:** The authors express their gratitude to the Secretaría Nacional de Ciencia, Tecnología e Innovación (SENACYT) of the Republic of Panama for its commitment and financial support towards the project. Special thanks are also due to the Master of Science in Mechanical Engineering program at the Faculty of Mechanical Engineering of the Universidad Tecnológica de Panamá. We further acknowledge the support of the Centro de Estudios Multidisciplinarios en Ciencias, Ingeniería y Tecnología AIP (CEMCIT-AIP) and The National Research System (Sistema Nacional de Investigación, SNI) of the Republic of Panama. The authors also thank the Laboratorio de Análisis Industriales y Ciencias Ambientales (LABAICA lab) at the UTP for FTIR analysis. Finally, we extend our gratitude to the Biosolids Laboratory at the Centro de Investigaciones Hidráulicas e Hidrotécnicas (CIHH) of the Universidad Tecnológica de Panamá for all the support provided during the development of the research.

**Conflicts of Interest:** The authors declare no conflicts of interest.

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
