# Peer review of "Enhanced Biological Nitrate Removal from Groundwater in Humid Tropical Regions Using Corn Cob-Based Permeable Reactive Barriers: A Case Study from Panama"

_water, doi:10.3390/w16121668_

Round 1
Reviewer 1 Report
Comments and Suggestions for Authors
Authors used agricultural residues, e.g., corn cob, to removal nitrate in groundwater. Four different concentrations were considered to understand the nitrate removal efficiency and denitrification rate. Additionally, the surface characteristics of corn cob were studied to assess the carbon sources stable and potential affect.
(1) Total nitrogen was quantified by summing the concentrations of nitrate, nitrite, and ammonium in this study. How to assess the organic nitrogen contribution during the batch test. Corn cob may leach the organic nitrogen and subsequent transfer to inorganic N.
(2) It did not show any discussion or analysis on the leaching assay in Table 2. This part can be deleted.
(3) It looks more different on NO2- and NH4 among the Figures 5 and 8, which may depend on the HRT. Authors should provide the HRT of Figure 5 and HRT in the natural groundwater, which may help to understand the potential denitrification in nature ecosystem.
(4) There are 13 figures. Some of them can be deleted or move to the supplementary information file.
(5) L711-718, All of them should be move to the figure caption.
(6) DQO in Figure and Horas in Figure 8 is not right presentation.
Author Response
Water, water 3034918
<< Enhanced biological nitrate removal from groundwater in Hu-mid Tropical regions using corn cob-based permeable reactive barriers: A case study from Panama >>
Dear editor and reviewers,
Thank you for your reply and the reviewers’ comments on our manuscript numbered “water-3034918”. Those comments are extremely valuable and helpful for the manuscript's revision and improvement. We have studied the comments carefully and have made corrections. Thank you very much for your consideration. Revisions are marked red in the manuscript. The responses to the reviewer's comments are as follows:
Responses to reviewer comments:
Note: This document is uploaded in a Word version, we realized that there were some errors in the application of the citations, so the entire document was reviewed and corrected.
Responses to comments will be made using the color red
Reviewer 1:
- Total nitrogen was quantified by summing the concentrations of nitrate, nitrite, and ammonium in this study. How to assess the organic nitrogen contribution during the batch test. Corn cob may leach the organic nitrogen and subsequent transfer to inorganic N.
Thank you for your valuable observations on our study. We understand the importance of evaluating the contribution of organic nitrogen during the batch test, especially considering that corn cob can leach organic nitrogen, which could then be transformed into inorganic forms of nitrogen.
In the leaching test, the contribution of inorganic nitrogen was determined by summing the leaching concentrations of nitrite, ammonium, and nitrate from the corn cob, so it was considered, based on the literature, that this contribution was found at low levels. In the characterization of the corn cob, 0.07% of nitrogen was identified in its elemental composition, which is utilized by denitrifying bacteria.
This study focused on quantifying the inorganic forms of nitrogen. Therefore, the evaluation of organic nitrogen was not included. However, it was considered in the elemental analysis of the corn cob and the inorganic nitrogen in the leaching test. Therefore, the analysis of organic nitrogen was not considered in the batch test. We appreciate the suggestions and will consider these recommendations to improve the quality and scope of our future research.
- It did not show any discussion or analysis on the leaching assay in Table 2. This part can be deleted.
We appreciate the reviewers' constructive comments. WE HAVE added a discussion about table 1 and table 2, in this case we could keep table 2 for a better understanding of the reader. In case they consider removing it again, we may remove Table 2 from the main manuscript and include it in the supplementary material (L336-L340)
- It looks more different on NO2- and NH4 among the Figures 5 and 8, which may depend on the HRT. Authors should provide the HRT of Figure 5 and HRT in the natural groundwater, which may help to understand the potential denitrification in nature ecosystem.
We appreciate the reviewer's comment. We acknowledge that the differences in NO2- and NH4+ concentrations between Figures 5 and 8 could be influenced by the HRT.
Given that HRT is an important factor in denitrification processes, we agree that its inclusion in our study would be beneficial for a better understanding of contaminant dynamics in the system. We have added information about the HRT analyzed in this study and its relation to the HRT of natural groundwater described in previous studies (L572-L584).
The batch test was conducted with the aim of analyzing the nitrate removal efficiency of corn cob at different concentrations, while the column test was conducted at different HRTs while maintaining an initial nitrate concentration.
However, as the figures may cause confusion at this point, we are considering the possibility of removing them to improve the clarity of the article. We believe this could enhance the understanding of the study.
- There are 13 figures. Some of them can be deleted or move to the supplementary information file.
Thank you for your suggestion. We have removed Figure 5 and Figure 8 from this manuscript and updated the numbering of the remaining figures.
- .L711-718,All of them should be move to the figure caption.
Thank you for your comment, we have corrected as suggested. The title of Figure 8 has been added (L730 – L742) and Figure 9 (L758 - L763).
- DQO in Figure and Horas in Figure 8 is not right presentation.
Thank you for your comment. We have removed the figure and reviewed the other figures to avoid making the same mistake.

Reviewer 2 Report
Comments and Suggestions for Authors
I have reviewed manuscript titled “Enhanced biological nitrate removal from groundwater in Humid Tropical regions using corn cob-based permeable reactive barriers: A case study from Panama". The material is presented appropriately and quite clearly, the data contained in figures represent understandable documentation of the research problem. The article has interesting results but a questions arose:
1) To what extent can such research be used in real conditions? Is it technically possible to create conditions similar to those in a laboratory in a natural environment?
2) according to table 1 Corn Cob contains high concentrations of iron and zinc. Do the Authors know to what extent the concentration of these elements in groundwater will increase? Have such studies been conducted?
Author Response
Water, water 3034918
<< Enhanced biological nitrate removal from groundwater in Hu-mid Tropical regions using corn cob-based permeable reactive barriers: A case study from Panama >>
Dear editor and reviewers,
Thank you for your reply and the reviewers’ comments on our manuscript numbered “water-3034918”. Those comments are extremely valuable and helpful for the manuscript's revision and improvement. We have studied the comments carefully and have made corrections. Thank you very much for your consideration. Revisions are marked red in the manuscript. The responses to the reviewer's comments are as follows:
Responses to reviewer comments:
Note: This document is uploaded in a Word version, we realized that there were some errors in the application of the citations, so the entire document was reviewed and corrected. Responses to comments will be made using the color red
Reviewer 2:
1) To what extent can such research be used in real conditions? Is it technically possible to create conditions similar to those in a laboratory in a natural environment?
We appreciate your valuable comments on our manuscript. Below, we provide a detailed response to your observation regarding the applicability of our research in natural conditions:
In our study, we aimed to evaluate different hydraulic retention times (HRT) that closely resemble soil conditions, considering their variability. Additionally, different nitrate concentrations were assessed. To simulate natural conditions, experiments were conducted in an environment that mimics a humid tropical climate. Corn cobs native to Panama were used, and no additional denitrifying bacteria were introduced in any of the experiments; therefore, the native bacteria from the corn cobs were sufficient for nitrate removal at various concentrations, ranging from 20 mg/L to 115 mg/L of NO3-N.
To establish the initial nitrate concentration, potassium nitrate fertilizer was used, which is commonly employed in agricultural activities and is the main cause of groundwater contamination by this pollutant. For this reason, we chose to use this source to test conditions as close to reality as possible.
In the column tests, covered columns were used to prevent solar radiation, and groundwater was used along with agricultural fertilizer as the nitrate source. No additional denitrifying bacteria were added; the bacteria present were native to the reactive material, in this case, corn cob.
We believe that the results obtained could be replicated or used as a reference for application in natural conditions. These findings provide a solid basis for conducting field pilot studies, where variables can be evaluated and adjusted according to the specific conditions of the site. Consequently, we believe that corn cob used as a reactive material in permeable reactive barriers has great potential for nitrate removal in groundwater in natural environments.
A paragraph has been added to expand this discussion (L229-L230), (L572-L584) in the revised manuscript. We thank you again for your suggestions and remain attentive to any other observations you may have.
- According to table 1 Corn Cob contains high concentrations of iron and zinc. Do the Authors know to what extent the concentration of these elements in groundwater will increase? Have such studies been conducted?
We appreciate your valuable comments on our manuscript. Below, we provide a detailed response to your observation regarding the potential release of metallic elements from corn cob into the environment.
The results of the leaching analysis indicated low concentrations of iron, copper, zinc, and sodium in the corn cob (Table 2). These findings suggest that although these elements are present in the corn cob (Table 1), their release into the environment through leaching is minimal. Therefore, it is unlikely that they pose a significant risk to groundwater. Xu et al. 2009 [1], indicated that metallic elements present in the leachate can have positive effects on denitrification rates, as metal ions are used as active centers in the denitrification process.
For greater reader comprehension, we have expanded the discussion in the manuscript (L336-L340) to include this information.
References
- Xu, Z.; Shao, L.; Yin, H.; Chu, H.; Yao, Y. Biological Denitrification Using Corncobs as a Carbon Source and Biofilm Carrier. Water Environment Research 2009, 81, 242–247, doi:10.2175/106143008x325683.
